# Quantifying congestion with player tracking data in Australian football

**Jeremy P. Alexander**[1]*, **Karl B. Jackson**[2], **Timothy Bedin**[2‡], **Matthew A. Gloster**[2‡], **Sam Robertson**[1]

**1** Institute for Health and Sport (iHeS), Victoria University, Melbourne, VIC, Australia, **2** Champion Data Pty Ltd, Melbourne, VIC, Australia

☯ These authors contributed equally to this work.
‡ TB and MAG also contributed equally to this work.
* jeremy.alexander@vu.edu.au

**Data Availability Statement:** All relevant data are within the manuscript and its Supporting Information files.

**Funding:** The authors Jeremy Alexander (JA), Karl Jackson (KJ), Timothy Bedin (TB), and Matthew

## Abstract

With 36 players on the field, congestion in Australian football is an important consideration in identifying passing capacity, assessing fan enjoyment, and evaluating the effect of rule changes. However, no current method of objectively measuring congestion has been reported. This study developed two methods to measure congestion in Australian football. The first continuously determined the number of players situated within various regions of density at successive time intervals during a match using density-based clustering to group players as 'primary', 'secondary', or 'outside'. The second method aimed to classify the level of congestion a player experiences (high, nearby, or low) when disposing of the ball using the Random Forest algorithm. Both approaches were developed using data from the 2019 and 2021 Australian Football League (AFL) regular seasons, considering contextual variables, such as field position and quarter. Player tracking data and match event data from professional male players were collected from 56 matches performed at a single stadium. The random forest model correctly classified disposals in high congestion (0.89 precision, 0.86 recall, 0.96 AUC) and low congestion (0.98 precision, 0.86 recall, 0.96 AUC) at a higher rate compared to disposals nearby congestion (0.72 precision, 0.88 recall, 0.88 AUC). Overall, both approaches enable a more efficient method to quantify the characteristics of congestion more effectively, thereby eliminating manual input from human coders and allowing for a future comparison between additional contextual variables, such as, seasons, rounds, and teams.

## Introduction

Australian football (AF) is a popular invasion sport played with two teams of 18 players on the field at any one time [1, 2]. Matches are divided into four quarters, each with 20 minutes of playing time [3]. The premier competition is the Australian Football League (AFL), which currently consists of 18 teams located across Australia [1]. Match-play has experienced a continual state of evolution, with improved player athleticism and professionalism, rule changes, innovative coaching tactics, and specialised training regimes all contributing to a faster game speed

Gloster (MG), are part-time or full-time employees of Champion Data. The funder provided support in the form of salaries for authors JA, KJ, TB, and MG but did not have any additional role in the study design, statistical analysis, decision to publish, or preparation of the manuscript. The specific roles of these authors are articulated in the 'author contributions' section.

**Competing interests:** The authors have declared that no competing interests exist. The affiliation of authors JA, KJ, TB, and MG to Champion Data does not alter the authors adherence to PLOS ONE policies on sharing data and materials.

[4, 5]. Contemporary AF has been described as largely defensive, whereby a combination of an increased number of tackles, contested possessions, stoppages in play, and a decrease in scoring and effective disposal rates, have been associated with greater player density around the ball [4, 6].

Given the potential negative implications this style of play may have on viewership and participation, various rule changes have been continually introduced by the AFL, with the intention of either negating or arresting the abovementioned trends [6]. Some recent modifications include enforcing an even spread of players across the field when a quarter begins or after a goal is scored and restricting opposition movement after a player marks the ball, to permit less restricted ball movement [7]. The introduction of these rule changes have typically aimed to reduce congestion and promote scoring by diminishing defensive strategies and stimulating more attacking styles of play through a more continuous free-flowing game [6, 7].

Preliminary investigations involving the evolution of match-play in AF promoted the notion that the speed of the game was increasing, which was estimated by measuring the average velocity of the ball in m·s −1 [5]. Specifically, game speed almost doubled in the Victorian Football League (VFL) and AFL, between the 1961 and 1997 seasons [5]. The overall trend of faster game speed continued until 2007, followed by a plateau, before a decrease through until 2015 [5]. This finding may correspond with teams allocating more emphasis on defensive actions and intentionally increasing player density around the ball, thereby increasing congestion [6, 8]. Consequently, opposition ball movement may be restricted by limiting the time and space afforded to opposing players [4, 5]. Congestion in AF has typically been inferred via video analysis, using a human coder to count the number of players within a five-metre radius of the ball at 15 s intervals [5]. It was revealed that congestion steadily increased through 2015, with 28.6% of time in-play witnessing at least five players being recorded within five metres of the ball, up nearly double since 2006 [5].

Nonetheless, a metric that provides a reliable and continuously measured description of congestion remains absent. Previous methods have been laborious, inefficient, and prone to error. In addition, an intermittent recording of the player count within a pre-determined region is inadequate to determine the comparative degree of congestion a player is confronted with when executing skilled actions, such as disposing of the ball. Whilst raw player counts can be used to infer congestion, the designated number of players as the threshold is somewhat arbitrary, which may render it challenging to agree on a widely accepted definition.

Considering the above, a reliable and valid method that can be scaled in an efficient manner would be useful. With the advent of player tracking technologies, a suitable data source is available, whereby the location of teammates and opponents at each point in time can be processed with machine learning algorithms to quantify the characteristics of congestion more effectively. Therefore, the aim of this study was to develop two methods to measure congestion in AF. The first continuously determined the number of players situated within various regions of density at successive time intervals during a match using clustering. The second determined the level of congestion a player experiences with when disposing of the ball using a classification approach. Both approaches were used to compare congestion across common AF contextual variables, such as field position and quarter.

## Materials and methods

### Data collection

Ethical clearance was granted by the University Human Research Ethics Committee (application number HRE20-172). Data were collected from the 2019 and 2021 AFL regular seasons and pooled for all analyses. Matches played in 2020 were not included due to the season

alterations that were implemented because of Covid-19. To ensure consistent tracking data and uniform field dimensions, matches ($n = 56$) were played at a single stadium (Marvel Stadium, Melbourne, Australia) where the field dimensions were 159.5 m x 128.8 m (length x width). Positional data in the form of Cartesian coordinates for each match were gathered using Catapult ClearSky 10 Hz local-positioning system (LPS) devices for all 44 participants (Catapult Sports, Melbourne, Australia). Teams were labelled Home Team and Away Team for each match to streamline data processing and visualisation. Matches were undertaken with four 20-min quarters (Q1, Q2, Q3, Q4) with breaks interspersed between periods. Tracking devices were housed in a sewn pocket in the jersey that is located on the upper back. Periods of play that lost the positioning of one or more players were omitted.

## Data analysis

Match event data were recorded by trained human operators to the nearest tenth of a second (Champion Data, Pty Ltd., Melbourne, Australia). This data provides information regarding players executing skilled actions, such as kicks, handballs, marks, where disposals are the total number of kicks and handballs. Previous investigations have assessed the validity and reliability of similar match event data and reported very high levels (ICC range = 0.947–1.000) of agreement [9]. Movement data derived from tracking devices were also recorded to the nearest tenth of a second and were synchronised with match event data using the unix timestamps present in both datasets [10]. This combined dataset was used to infer the location of the ball, which was also specified to the nearest tenth of a second. Field position of the ball was separated into four zones (defensive 50; D50, defensive midfield; DM, attacking midfield; AM, forward 50; F50) by the two 50 m arcs and the centre of the ground, which is orthodox for AF research and statistical providers [11–13]. Periods where the ball was out of play, for example, when there was a break between quarters, when the umpire had the ball before a stoppage, and after scores were excluded from the investigation [14].

## Continuous congestion during match play

The proposed concept of measuring continuous congestion is to differentiate between higher player density and lower player density at each successive time interval during a match. Clustering is ideally suited to this proposition due to the capacity of partitioning data into groups based on the similarities of their properties [15]. Depending on the state of play at any one time, most of players could be positioned in a single region of the ground, producing one large cluster of congestion, or separated in multiple distinct groups, generating several smaller clusters of congestion, or evenly spaced across a field of a play with no observable congestion. Consequently, is it necessary that any potential clustering technique involves a flexible mechanism that can manage a variable number of input groups, rather than a strict assignment of players to a pre-determined fixed number of groups.

Density-based clustering techniques, such as Density-Based Spatial Clustering of Applications with Noise (DBSCAN) and Ordering Points to Identify the Clustering Structure (OPTICS), satisfy the aforementioned impediments to quantify congestion. Specifically, these algorithms are established on reachability, whereby, data is clustered by identifying a minimum number of points within the neighbourhood of a certain radial threshold [16, 17]. Points that meet this criterion are considered 'core points', whilst data not meeting this criterion are referenced as 'noise' [18]. The OPTICS algorithm is preferable to DBSCAN as it has the capacity to detect meaningful clusters of varying density by ordering the representation such that points that are spatially nearest become neighbours and it only requires the minimum number

of points as a mandatory input parameter [15]. As such, the OPTICS model was used to differentiate players within regions of higher density compared to those of lower density.

## Analysis for continuous congestion during match play

OPTICS clustering model was performed using the *scikit-learn* library in *python* [19]. Players were clustered at each successive time interval during a match, whereby all 36 players were clustered as either core points or noise based on their respective location. To be considered a core point (*core−distance*), a player *p* had to be within a maximum radius $\varepsilon$ of 7.5 m and contain a minimum number of three additional players (*MinPts−distance*) within its $\varepsilon$−neighbourhood $N_{\varepsilon}(p)$. The *core−distance* of a player is the smallest distance between another player within its neighbourhood, meaning that player will become a core point if separate players are contained in its neighbourhood. Otherwise *core−distance* is *UNDEFINED*.

$$core-distance_{\varepsilon,MinPts}(p) = \left\{ \begin{array}{l} UNDEFINED, \; if \; |N_{\varepsilon}(p)| < MinPts \\ MinPts-distance(p), \; otherwise \end{array} \right\}$$

The *reachability−distance* of another player *o* from *p* is either the distance between *o* and *p*, or the *core−distance* of *p*, whichever is larger.

$$reachability-distance_{\varepsilon,MinPts}(o,p) = \left\{ \begin{array}{l} UNDEFINED, \; if \; |N_{\varepsilon}(o)| < MinPts \\ \max(core-distance(o), \; distance(o,p)), \; otherwise \end{array} \right\}$$

Players were considered to be within congestion if identified as core points, while players were deemed outside congestion if recorded as noise. Clusters of congestion were originally assigned output labels of 0-*n*, while -1 was assigned to all points clustered as noise (see Fig 1) [20, 21]. These labels where converted to provide a practical description of congestion and to differentiate between separate clusters of congestion if more than one cluster was identified for a unique time interval (see Fig 1). Specifically, the cluster of congestion with the highest count of players was re-labelled 'primary' congestion, while remaining cluster(s) containing a lesser number of players were re-assigned as 'secondary' congestion. Finally, -1 was converted to 'outside' congestion.

The OPTICS model was iterated on each individual time point in every match to assess the proportion of players within each cluster for each unique point in time. Descriptive statistics (mean ± standard deviation) for the proportion of players within each cluster from each match was compared across field position (see Fig 2) and quarter (see Fig 3).

## Classifying level of congestion during disposals

Whilst the aforementioned clustering model is able to group players inside congestion compared to those outside congestion, it is unsuitable to categorise the level of congestion a player is confronted with when disposing of the ball. Simply, as the output is reduced to either 'inside' or 'outside' congestion, it disregards a more specific account of congestion that describes the level of congestion experienced by the ball-carrier. As such, a supervised machine learning algorithm is preferable for this task, whereby the level of congestion can be classified from an input vector of variables. Specifically, a ground truth label can be ascribed to a set of disposals, from which a classifier model attempts to predict the same labels using an optimal set of features and parameters [22]. To provide a ground truth of the level of congestion a player experiences when disposing of the ball, a training dataset was generated by professional analysts from Champion Data, whereby 1943 disposals were manually labelled using three categories outlined in Table 1.

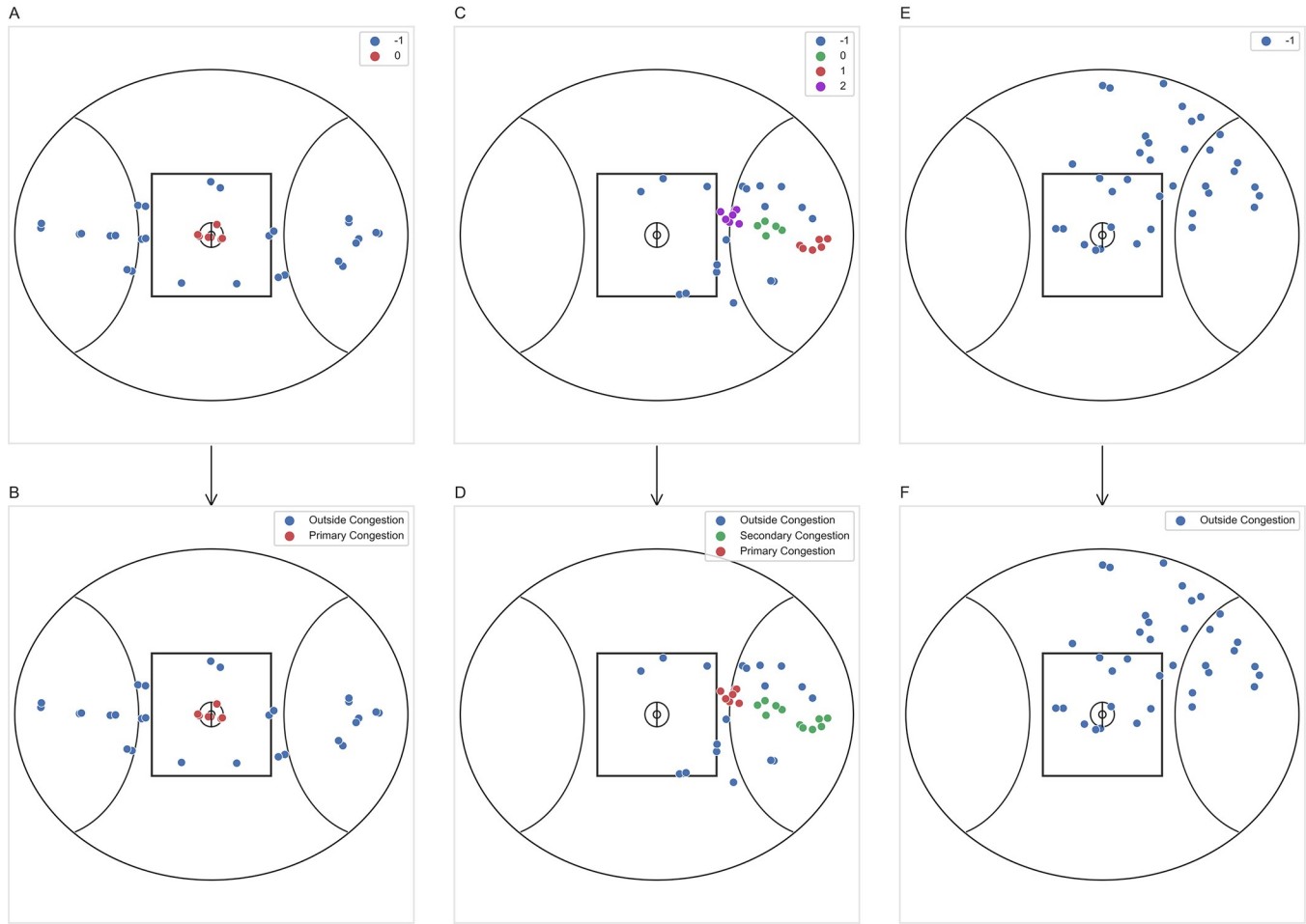

**Fig 1. Players clustered as primary, secondary, or outside congestion during three separate time intervals during a match.** Each column of subplots (**A-B, C-D, E-F**) are pairs representing the same time interval. Upper panel (**A, C, E**) represents initial output labels from the clustering model. Lower panel (**B, D, F**) displays the corresponding labels after translating into primary, secondary, and outside labels.

Determining an appropriate set of features to train a given model depends on technical comprehension, prior knowledge of the problem, or the purpose of the analysis [23]. In consultation with the same match analysts from Champion Data, a range of spatiotemporal features (Table 2) were developed. These features were assessed for every disposal in the dataset, which delivered information from which a model could classify the aforementioned level of congestion.

## Analysis for classifying level of congestion during disposals

All analyses were performed using the Scikit-learn library in *python*. To select the appropriate classification model, base model testing was run using the *lazypredict* package, which is a repository of classifier algorithms [24]. The classifier that yielded the highest accuracy was the Random Forest (RF) algorithm. The RF classifier is a non-linear machine learning technique used for classification and regression, whereby an assembly of decision trees are used to calculate the mode of classes of individual trees and ranking of classifiers [25]. The RF classifier was used to assess the disposal labels (High Congestion, Nearby Congestion, Low Congestion) when referencing the spatiotemporal features.

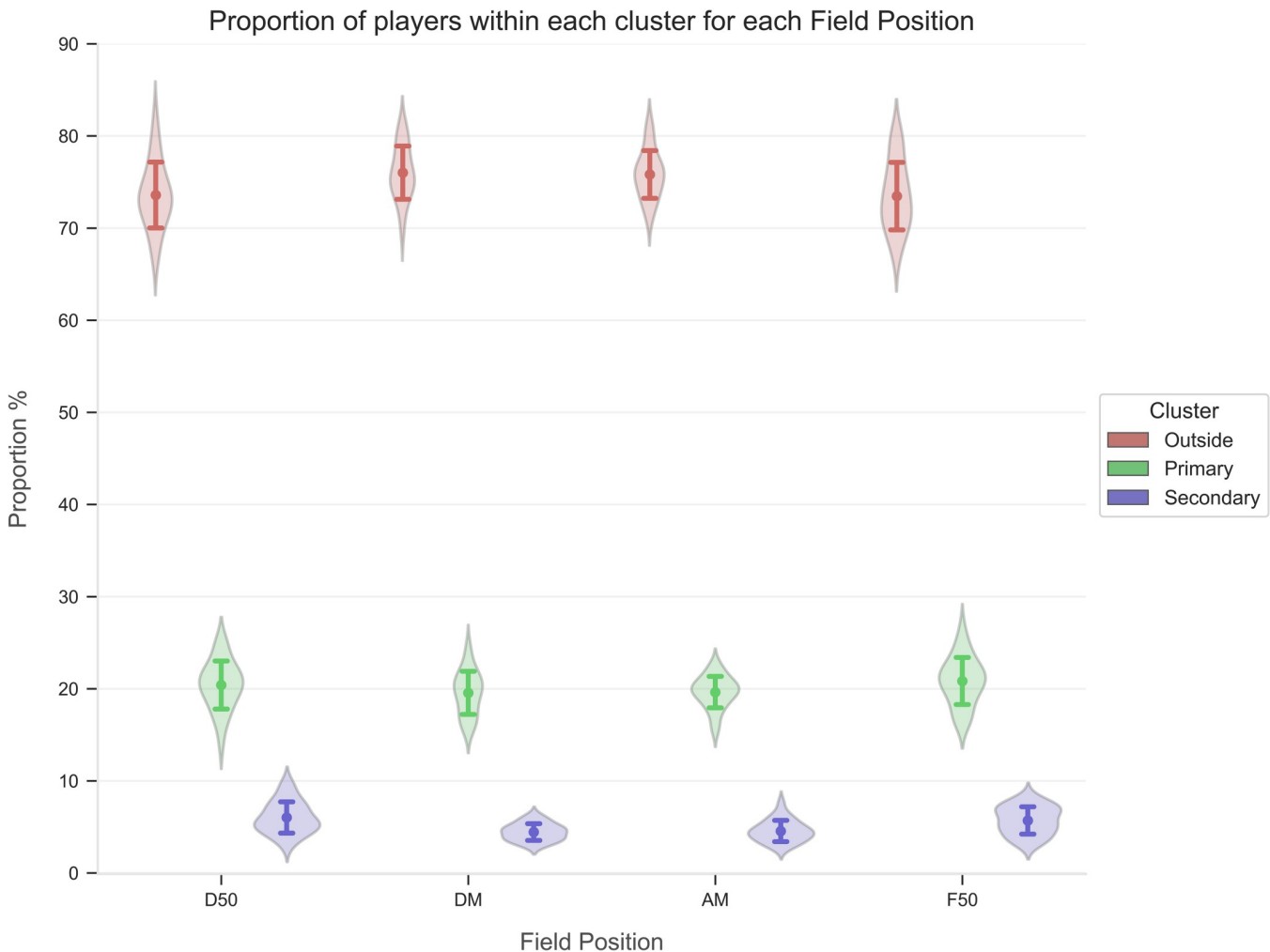

**Fig 2. Proportion of players (mean ± standard deviation) in each cluster compared across field position.** D50, Defensive 50; DM, Defensive Midfield; AM, Attacking Midfield; F50, Forward 50.

The data were split into training and testing (80:20) datasets [26, 27]. The RF classifier's hyperparameters were optimised using *GridSearchCV* in *scikit-learn* [28]. After fine-tuning for optimal performance, we selected the *Gini Index* for the *criterion*, the *number of trees* was fixed to 500, the *maximum depth* of each tree was set to 10, the *minimum sample split* was adjusted to 50, and the *minimum sample leaf* was set to 5. The feature importance scores were determined by the *Gini Index*, where feature extraction followed using the lowest contributing value out of each iteration until a decrease in model performance occurred [27], resulting in the removal of the 'available space' feature. The remaining 8 features were used for modelling.

To visualise and interpret the feature importance, the *Shapley Additive exPlanations* (SHAP) package was used [29]. This package displays the global importance of each feature for classifying the disposal label and the local explanation of each feature exhibiting the direction of the relationship between the feature and disposal label [29]. Model performance was assessed based on standard metrics including precision (ratio of true positives to predicted positives), recall (ratio of true positives to actual positives) and F1-Score (harmonic mean of precision and recall) [26]. The confusion matrix, ROC curves (receiving operating characteristics), and precision-recall (PR) curves were also examined to determine the performance of the

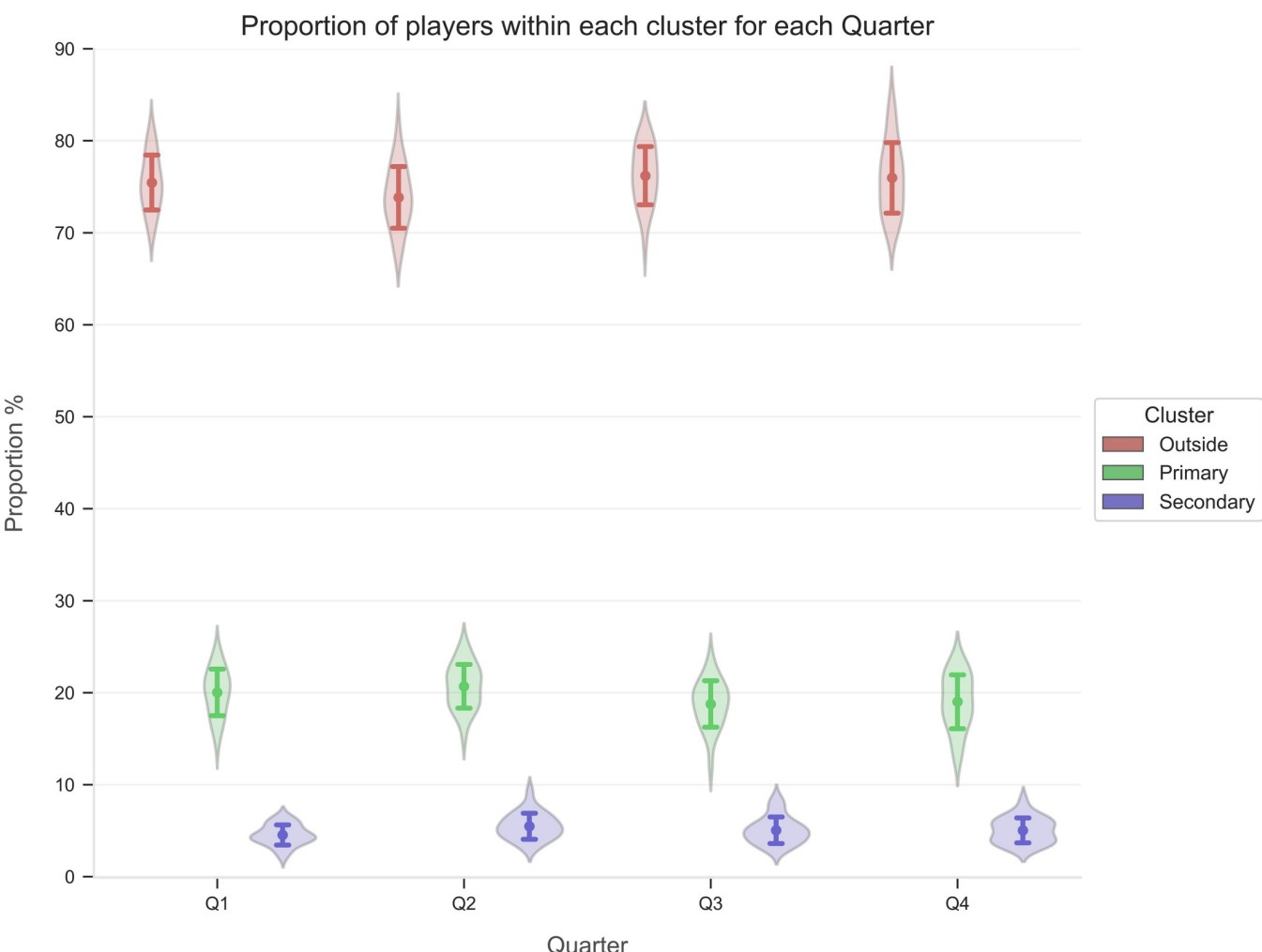

**Fig 3. Proportion of players (mean ± standard deviation) in each cluster compared across quarter.** Q1, Quarter 1; Q2, Quarter 2; Q3, Quarter 3; Q4, Quarter 4.

model [30]. Although ROC curves are typically used for binary classification, they can be administered to multi-class classification by using the one vs all approach and considering the micro-average curve to analyse the overall performance of the classifier [31]. The ROC curve is generated by plotting the false positive rate against the true positive rate [30]. The area under the ROC curve (ROC-AUC) describes the capacity to distinguish between classes, with an ROC-AUC of 1.0 representing that the classifier can differentiate classes perfectly [32]. The area under the precision-recall curve (PR-AUC) is produced by plotting the recall against the precision, which provides an indication of the number of positives samples in a dataset [33].

**Table 1. Definition of the level of congestion when disposing of the ball.**

| Disposal label | Description |
|---|---|
| **High Congestion** | Several players within 0–5 m of the ball-carrier |
| **Nearby Congestion** | Multiple players with 0–10 m of the ball-carrier but there is some space to make a decision |
| **Low Congestion** | There is one or no players within 10 m of the ball-carrier |

**Table 2. Definition of spatiotemporal features for disposal classification model.**

| Features | Description |
|---|---|
| Immediate Player Count (IPC) | Total count of all players within a 5 m radius of the ball-carrier |
| Extended Player Count (EPC) | Total count of all players within a 10 m radius of the ball-carrier |
| Immediate Defenders (IDC) | Total count of defenders within a 5 m radius of the ball-carrier |
| Extended Defenders (EDC) | Total count of defenders within a 10 m radius of the ball-carrier |
| Frontal Player Count (FPC) | Total count of players within frontal 90-degree quadrant of the ball-carrier |
| Right Player Count (RPC) | Total count of players within right 90-degree quadrant of the ball-carrier |
| Left Player Count (LPC) | Total count of players within left 90-degree quadrant of the ball-carrier |
| Behind Player Count (BPC) | Total count of players within behind 90-degree quadrant of the ball-carrier |
| Available space (AS) | Total area that intersects between radius that surrounds player/ball and the field of play |

The resulting RF classifier computed the label of every disposal in the dataset. Descriptive statistics (mean ± standard deviation) for each disposal label from each match determined the breakdown of the level of congestion during disposals, compared across field position and quarter.

## Results

### Continuous congestion during match play

Results from the OPTICS clustering model are presented in Figs 2 and 3. When assessing field position, the proportion of players in primary and secondary congestion was marginally greater in the D50 and F50 when compared to the DM and AM (Fig 2). Conversely, outside congestion witnessed a greater proportion of players in the AM and DM when compared to the D50 and F50. Finally, as the match progressed across each quarter, the proportion of players within primary congestion observed a minor decrease, while the segment of players outside congestion slightly increased (Fig 3).

### Classifying level of congestion during disposals

Fig 4 displays the global feature importance and the local explanation summary exhibiting the direction of the relationship between each feature and disposal label. Immediate player count, extended player count, and immediate defender count were the most important features in classifying the disposal label. Evaluation of the RF classifier is presented in Figs 5 and 6, using metrics, confusion matrix, ROC-AUC and PR-AUC curves for each disposal label. Disposals within high congestion, 0.89 precision and 0.86 recall, and low congestion, 0.98 precision and 0.86 recall, were correctly classified at a considerably higher rate compared to disposals nearby congestion, 0.72 precision and 0.88 recall. The ROC-AUC of 0.96 for disposals in high congestion and 0.96 for low congestion were greater than 0.88 in disposals nearby congestion. Similarly, the PR-AUC of 0.9 and 0.96 for disposals in high congestion and low congestion, were greater than 0.69 for disposals nearby congestion.

The breakdown of the level of congestion for each disposal label compared across field position and quarter are presented in Figs 7 and 8. When assessing field position (Fig 7), the proportion of disposals in high congestion increased as the ball transitioned from D50 to F50. Disposals nearby congestion increased from D50 to AM, while those outside congestion concurrently decreased. Conversely, progressing from the AM to F50 observed a decrease in disposals nearby congestion and an increase in disposals outside congestion. As a match

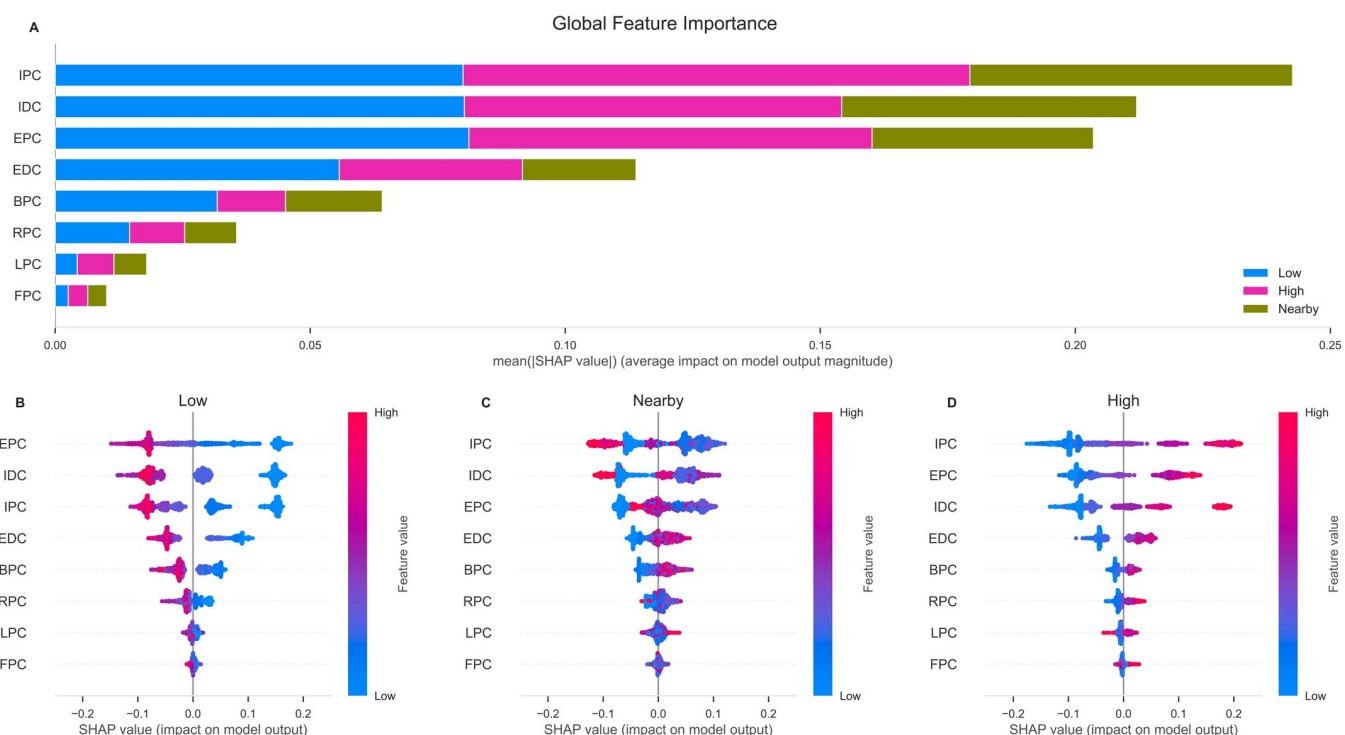

**Fig 4. Global importance and local importance of each feature using SHAP values.** Upper panel displays the global feature importance expressing the order of most important (top) to least important (bottom) of every feature to classify each respective disposal label (**A**). Lower panel represents the local explanation exhibiting the direction of the relationship between each feature and low congestion (**B**), nearby congestion (**C**), and high congestion (**D**) in the training dataset. The higher the local SHAP value of a feature, the higher the log odds of classifying the respective disposal label. The letters denote spatiotemporal features: IPC = Immediate Player Count; EPC = Extended Player Count; IDC = Immediate Defenders; EDC = Extended Defenders; BPC = Behind Player Count; RPC = Right Player Count; LPC = Left Player Count; FPC = Frontal Player Count.

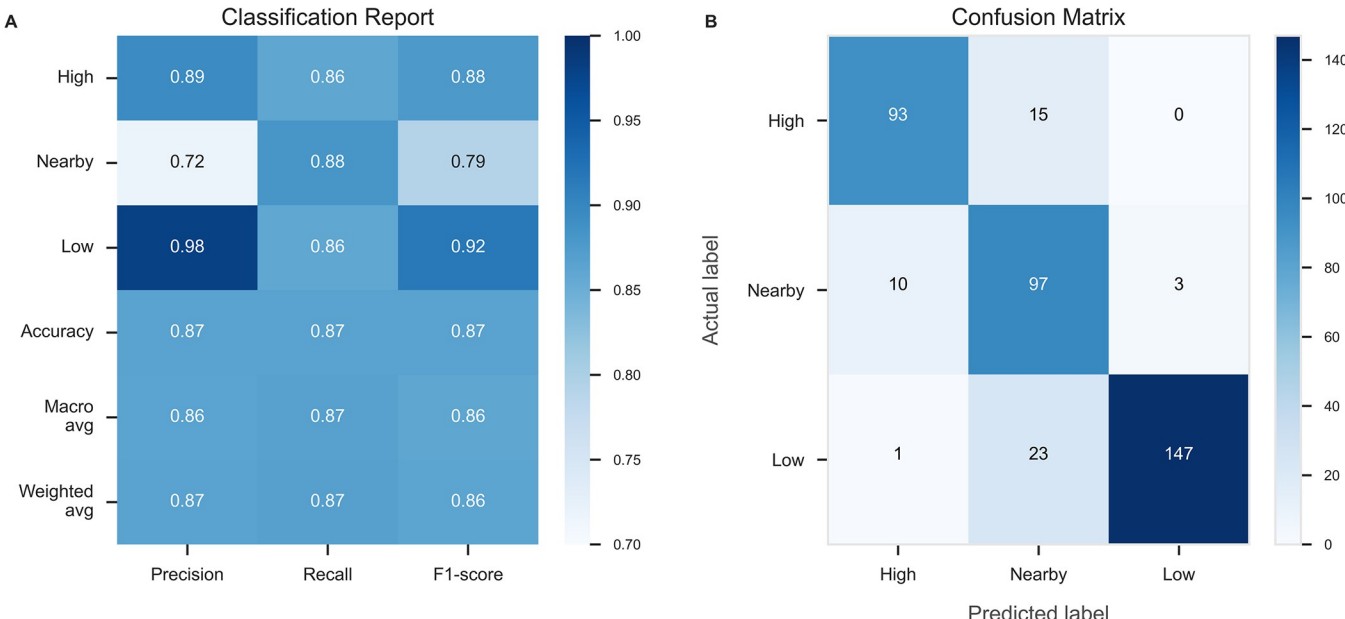

**Fig 5.** Evaluation metrics, including precision, recall, and F1-score assessing the performance of the RF model to classify each disposal label (**A**). Confusion matrix of the RF model displaying correctly classified and misclassified disposal labels (**B**).

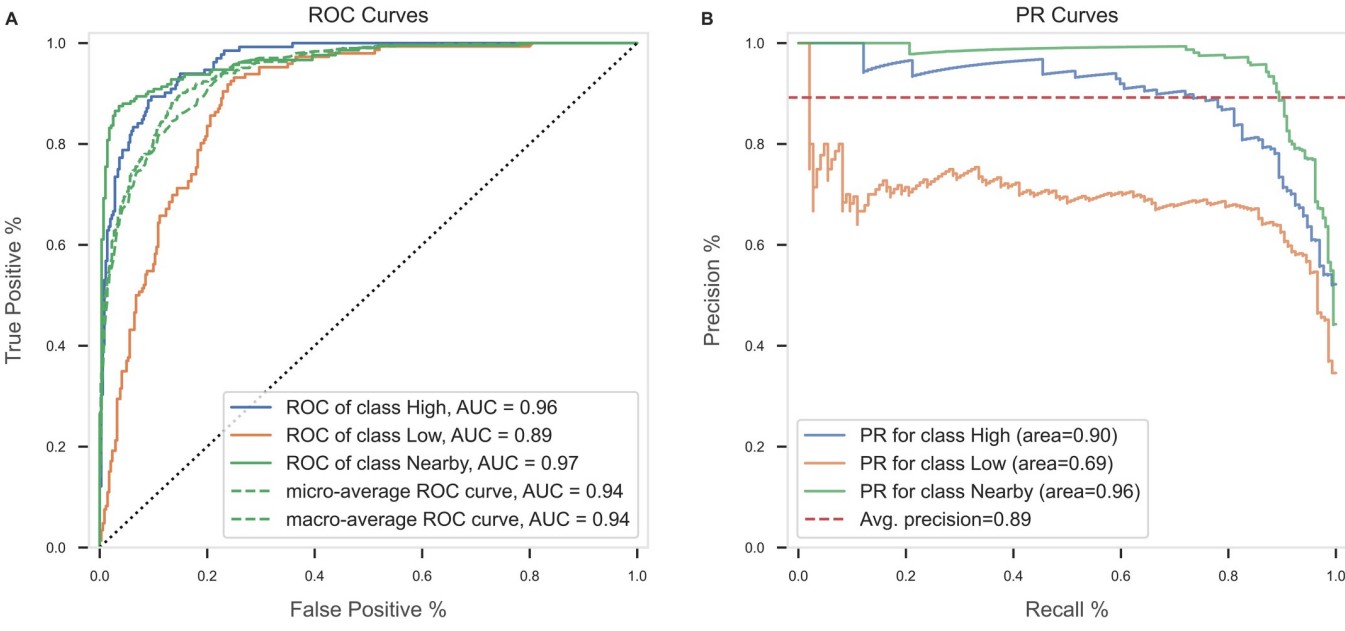

**Fig 6.** Evaluation of RF model to classify each disposal label expressed by ROC curves (**A**), and PR curves (**B**).

proceeded across each quarter (Fig 8), disposals within high congestion steadily decreased, while disposals nearby congestion and outside congestion gradually increased.

## Discussion

This study developed two methods to measure congestion in AF. The first continuously determined the number of players situated within various regions of density at successive time intervals during a match, whilst the second classified the level of congestion a player experiences when disposing of the ball. This information provides a scalable method to quantify congestion during matches.

The first method showed that players are within a cluster of congestion (primary or secondary) between 23% and 26% of a typical game. Whilst an exact comparison to existing research is challenging, given the differences in methodology, previous studies report similar findings in AF [5]. Specifically, 28.6% of total time in-play witnessed at least five players within 5 m of the ball during 15 second intervals in the 2015 season. This finding was more than double that of the 2006 season, which recorded 11.2% of time in-play.

Whilst a continuous account of congestion provides an indication of players located within clusters of greater density across a field of play, it is unsuitable to quantify the level of congestion a player experiences when disposing of the ball. Specifically, the output description is limited to 'inside' or 'outside' congestion, thereby excluding a more nuanced or tiered description of congestion experienced by the ball-carrier. In addressing these limitations, the RF model was able to correctly classify disposals within high congestion and low congestion at a higher rate when compared to disposals nearby congestion. These findings may be attributed to the fluid nature of congestion, whereby certain disposals may span across two separate categories, rather than neatly fit into a single category. For example, a disposal classified as nearby congestion may contain characteristics that are similar to either high congestion or low congestion, as is evident when considering the false positives in the confusion matrix. Overall, the model

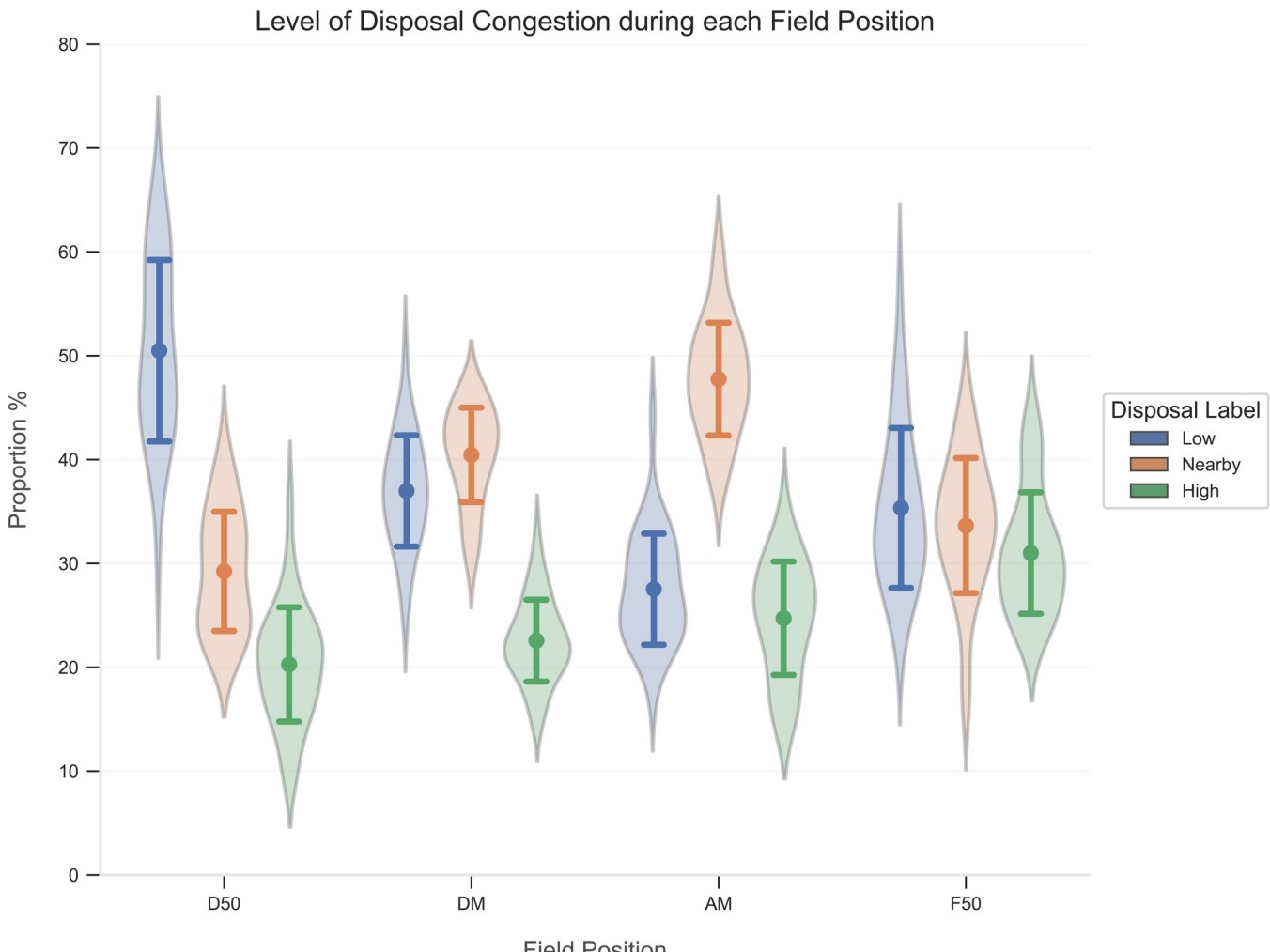

**Fig 7. Breakdown of disposals (mean ± standard deviation) in each level of congestion compared across field position.** D50, Defensive 50; DM, Defensive Midfield; AM, Attacking Midfield; F50, Forward 50.

eliminates the task of manually coding each unique label, thereby establishing a scalable method to quantify the level of congestion a player experiences when disposing of the ball.

Overall, more than 60% of disposals encountered high congestion or nearby congestion. This suggests that large segments of match-play experiences greater density around the ball-carrier, which may instil pressure and influence passing capacity. Previous research confirms comparable sentiments in AF, with effective disposal rates steadily declining since 2005, coinciding with a concomitant increase in the number of tackles [4, 34]. Disposals performed under low congestion decreased as teams transitioned the ball towards their attacking end. After a team obtains possession of the ball in the defensive half, the opposition may fold back in numbers to establish defensive stability, rather than press up the field to instigate a turnover in possession. Corresponding studies corroborate this tactical team behaviour in AF, whereby teams produced a numerical advantage in their defensive half [35]. Although disposals in low congestion decreased in the F50 compared to the AM, this is likely due to set shots on goal. Under this scenario, opposition players are prohibited from entering a designated space around the player, thereby allowing a shot at goal unimpeded by the opposing team. The level

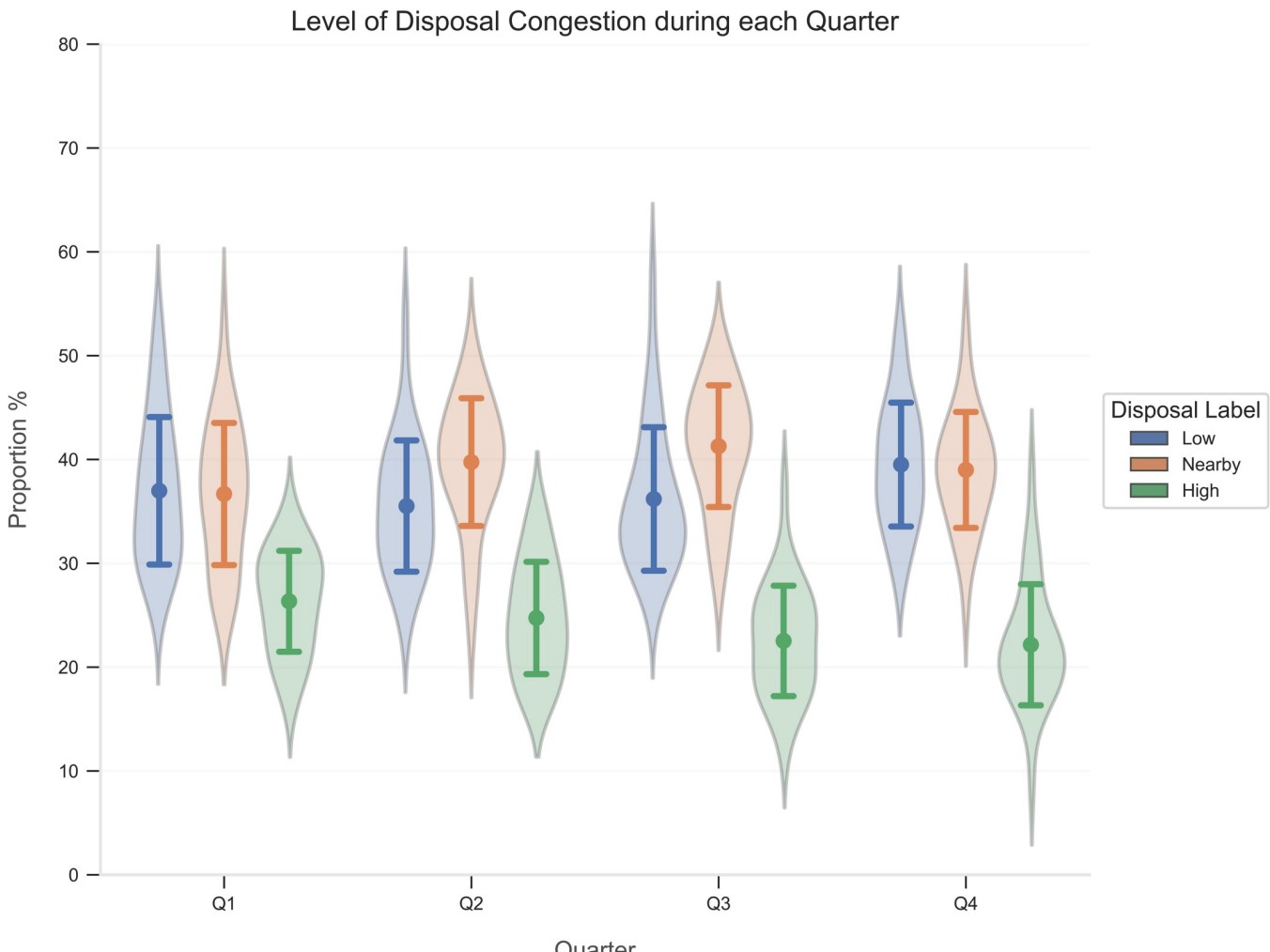

**Fig 8. Breakdown of disposals (mean ± standard deviation) in each level of congestion compared across quarter in the 2019 and 2021 season.** Q1, Quarter 1; Q2, Quarter 2; Q3, Quarter 3; Q4, Quarter 4.

of congestion steadily declined across each quarter, which may suggest that as time in play elapses, players fatigue or increased scoring margins witness a decrease in intensity or effort as the match outcome is largely determined.

In response to a steadily declining scoring rate and a predominantly defensive game style, the AFL, guided by the Laws of the Game Charter, have continually implemented major rule changes to enhance fan enjoyment [6, 8]. Initially, rule changes involved capping interchange rotations and initiating quicker re-starts to play after a score or the ball going out of bounds [8]. More recently, teams are required to maintain even numbers in each section of the ground at the commencement of each quarter and after a goal is scored [7]. In addition, all opposing players are prohibited from entering a designated region surrounding a player that obtained a mark, whilst compelling the player standing the mark to remain stationary [7]. Ostensibly, such rule changes constrain the defending team's ability to restrict ball transition, thereby allowing for more attacking ball movement for the offensive team, which may increase scoring [7]. However, scoring rates have remained stubbornly subdued [36], which suggests that the current rule changes may not be enticing teams to alter their collective movement behaviour.

Indeed, previous studies support this sentiment in AF, whereby teams are preserving a numerical advantage in their defensive 50, which reduces the likelihood of generating a shot on goal [35]. The direct causes of reduced scoring rates are likely multifaceted and require further investigation.

The trend towards a greater emphasis on defensive strategies and lower scoring rates has been reported in other invasion sports including football, rugby league, and rugby union [37, 38]. A key component of this development may be increased player density and congestion [38]. Nonetheless, limited work has been undertaken in measuring player congestion, except for techniques that involve considerable human input. Both approaches developed in this study demonstrate a scalable method to quantify congestion during match-play that require minimal manual control. This information can be applied to various aspects of performance analysis in invasion sports, such as, evaluating the efficacy of training programmes, assessing the physical demands of sport performance, quantifying the value of passes, and informing expected goals metrics. Specifically, sport science practitioners may incorporate a more representative training design by targeting drills that replicate congestion witnessed in match-play. The frequency of passing in football has increased in recent years [37]. Successful teams also record greater possession rates and an increased number of passes per game compared to their losing counterparts [39]. As player proximity is a central component in skill execution and attaining a goal [40], incorporating congestion may provide an enhanced understanding when quantifying the value of passes and expected goals metrics.

Whilst the dataset included in this investigation accounts for 56 matches across multiple seasons, it was limited to matches played at a single stadium. The inclusion of additional data may identify a more nuanced representation of congestion and if any variations exist between teams, stadiums, and independent rounds. The machine learning models proposed in this study to quantify congestion were novel, which naturally specifies the parameters used to tune the algorithms were likewise exploratory. Although the models were thoroughly trailed and tested using various input parameters, a greater implementation from a broader range of experts may assist in ensuring the methodology is valid and reliable and if alterations need to be tailored for specific applications. Additional investigations may also determine how congestion located elsewhere on the field influences subsequent match events. Specifically, determining how greater congestion forward of the ball influences match event outcomes, such as retaining possession or scoring.

## Conclusion

This study developed two methods for measuring congestion in AF. The clustering model identified that players were within a cluster of congestion between 23% and 26% of a typical game. The random forest model was able to correctly classify disposals in high congestion and low congestion at a higher rate compared to disposals nearby congestion. Both modelling approaches demonstrate a more efficient method to quantify the characteristics of congestion, thereby eliminating manual input. This information provides a scalable method to quantify congestion, which allows for the comparison between seasons, rounds, and teams and can be used to inform player training, team strategy and rule changes.

## Supporting information

**S1 File.**
(ZIP)

## Author Contributions

**Conceptualization:** Jeremy P. Alexander, Karl B. Jackson, Sam Robertson.

**Data curation:** Jeremy P. Alexander, Timothy Bedin, Matthew A. Gloster.

**Formal analysis:** Jeremy P. Alexander.

**Methodology:** Jeremy P. Alexander, Sam Robertson.

**Supervision:** Karl B. Jackson.

**Visualization:** Jeremy P. Alexander.

**Writing – original draft:** Jeremy P. Alexander.

**Writing – review & editing:** Jeremy P. Alexander, Karl B. Jackson, Timothy Bedin, Matthew A. Gloster, Sam Robertson.

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
