## [Decision Letter · Decision Letter 0]

21 Apr 2022

PONE-D-22-01015Quantifying congestion with player tracking data in Australian FootballPLOS ONE

Dear Dr. Alexander,

Thank you for submitting your manuscript to PLOS ONE. After careful consideration, we feel that it has merit but does not fully meet PLOS ONE’s publication criteria as it currently stands. Therefore, we invite you to submit a revised version of the manuscript that addresses the points raised during the review process.

 Please submit your revised manuscript by Jun 05 2022 11:59PM. If you will need more time than this to complete your revisions, please reply to this message or contact the journal office at plosone@plos.org. Please include the following items when submitting your revised manuscript:A rebuttal letter that responds to each point raised by the academic editor and reviewer(s). You should upload this letter as a separate file labeled 'Response to Reviewers'.A marked-up copy of your manuscript that highlights changes made to the original version. You should upload this as a separate file labeled 'Revised Manuscript with Track Changes'.An unmarked version of your revised paper without tracked changes. You should upload this as a separate file labeled 'Manuscript'.If applicable, we recommend that you deposit your laboratory protocols in protocols.io to enhance the reproducibility of your results. Protocols.io assigns your protocol its own identifier (DOI) so that it can be cited independently in the future. For instructions see: https://journals.plos.org/plosone/s/submission-guidelines#loc-laboratory-protocols. Additionally, PLOS ONE offers an option for publishing peer-reviewed Lab Protocol articles, which describe protocols hosted on protocols.io. Read more information on sharing protocols at https://plos.org/protocols?utm_medium=editorial-email&utm_source=authorletters&utm_campaign=protocols.

We look forward to receiving your revised manuscript.

Kind regards,

Gábor Vattay, PhD, DSc

Academic Editor

PLOS ONE

Journal Requirements:

2. Please provide additional details regarding participant consent. In the ethics statement in the Methods and online submission information, please ensure that you have specified (1) whether consent was informed and (2) what type you obtained (for instance, written or verbal, and if verbal, how it was documented and witnessed). If your study included minors, state whether you obtained consent from parents or guardians. If the need for consent was waived by the ethics committee, please include this information

Reviewers' comments:

Reviewer's Responses to Questions

**Comments to the Author**

1. Is the manuscript technically sound, and do the data support the conclusions?

Reviewer #1: Yes

Reviewer #2: Yes

2. Has the statistical analysis been performed appropriately and rigorously? 

Reviewer #1: Yes

Reviewer #2: Yes

3. Have the authors made all data underlying the findings in their manuscript fully available?

Reviewer #1: Yes

Reviewer #2: Yes

4. Is the manuscript presented in an intelligible fashion and written in standard English?

Reviewer #1: Yes

Reviewer #2: Yes

5. Review Comments to the Author

Reviewer #1: Comments to the authors

General comments

Thanks for allowing me to review this manuscript. Overall, the study is of very high quality and the findings are applicable and relevant to those working within AFL. The two methods presented to quantify congestion are novel and highly innovative, requiring extensive analyses so I commend you all on that. A major strength of the study was the ability of the authors to explain the findings in simple, practical terms, useful for practitioners as the analyses would be unfamiliar to most readers. Although I have some understanding of machine learning, much of the analyses were unfamiliar to me, so I can’t comment too much about the appropriateness, however the figures were easy to interpret. The manuscript is very well written, and explains the concept of the study exceptionally, and the as mentioned the findings are explained well. My suggestions/comments are mostly related to some additional detail that I believe is necessary within the methods which are detailed below. Thanks again, and congrats on completing such a high quality manuscript.

Specific comments

Abstract

Line 40: Consider adding ‘method’ between second and aimed. To me it read like an aim of the study, when really you’re referring to the method

Introduction

Overall, the introduction is really well written and provides context to the reader about the purpose of the study

Line 75-76: Where you refer to the ‘speed’ of the game, you need a bit more detail around this. Specifically, are you referring to m/min (assuming so), but I can imagine many readers may assume speed as in velocity of running and perhaps high-speed running volumes. Provide descriptive data in line 77.

Methods

Line 109: Why was 2020 not included? Was this due to the reduced duration of games?

Line 112: Did you obtain data from all teams in the AFL or one team? How did you obtain this data if it involved multiple teams?

Line 113: The S5 devices aren’t LPS enabled, these devices would have been worn in the 2019 season, but Vector would have been used in 2020-2021. In 2019, Catapult Clearsky (LPS devices) were worn in Marvel stadium. Can you confirm this information, as well as detail/report any between-device information that’s necessary here.

Line 133: Please provide detail on your methods of determining player location data. I’m assuming you used GNSS lat/long data, but please include details on this within the methods

Figure 1: The quality of the figure seems quite poor in the PDF. I can see it’s a tif file, but perhaps check it in the next submission. Also, I wonder if it’s necessary to include the top panel here – for simplicity?

Line 193: State where that data is presented – Figure 2 and 3

Line 214: Did the same Champion Data staff label/code the disposals?

Line 233: Should the Shapely Additive exPlanations package be in italics like the others previously reported?

Results

Overall, the results explain the findings really well. My only suggestion would be with the figures to make them look a bit ‘cleaner’ and publication worthy. For example, removing the grey section around them, lighten (or remove) the gridlines, position the legend in a consistent spot. Also the colour scheme used across each figure varies quite a bit. I understand the colours are used to represent different things, but perhaps consider using a similar theme

Discussion

The discussion was well written, and explained the results in simple, applied/practical terms. This is really useful for readers, as the analysis is very complex, the findings need to be interpreted clearly for the translation of this study into practice.

Line 314: Perhaps link the first two paragraphs together.

Line 382: Can you state the reasoning for this in the methods

References

Ref 2, 14, 34, 35 journal title needs to be in capital format

Reviewer #2: The authors are to be commended for a well-written article. It is the first known article that proposes objective data analysis techniques to capture the congestion in Australian football. Both approaches seem to provide more effective information than current methods (i.e., manual input). This not only seems to be useful to evaluate congestion in AF, but also may be practical for many other collective sports-related phenomena. Generally, the topic falls within the scope of the journal and could be of potential interest to its readers.

However, the following minor concerns need to be addressed before publication:

Due to the large amount of abbreviations a list of abbreviations would be recommended, as long as journal’s guidelines approve it, to facilitate the reading.

Abstract:

Ln34: Australian instead of Australia

*Ln35: “is an important consideration in identifying passing capacity, assessing fan enjoyment, and evaluating the effect of rule changes”. You mention these aspects as important when studying congestion but they are not mentioned elsewhere in the article aside from the abstract. Why? In my opinion, they should be mentioned if you consider in the introduction or discussion, otherwise I would remove them from the abstract.

Ln54: Congestion is already in the title. I would suggest to switch it for another key word to avoid repeat it. In this way it may facilitate this article to be found from more searches.

Materials and methods:

*Ln203 to Ln214: As I see, the definitions of level of congestion and spatiotemporal features for disposal classification model are done by yourself without any evidence but in consultation with professionals. I understand there is no work that establishes previous criteria about it. But why these levels and not others?

As you may understand these definitions may be somehow relative to the individual and environmental constraints of the game. For example, for nearby congestion “multiple players with 0-10m of ball-carrier but there is some space to make a decision”: within this 10 m some experienced and fresh player probably will have time to make a clear decision, however, a novice and fatigued young player during a rainy day probably can feel high congestion in this space to make an “adequate” decision. I am not intended to change your definitions but to help future readers to consider different features for classifying the dynamic and nonlinear level of congestion, assessing previously the main constraints of the game (e.g., age of players, level of players, meteorology, etc.). In my opinion, this should be briefly considered as a practical implication (in discussion) for next studies to do not treat it as a universal and fixed rule.

Discussion:

Ln310: Why do you not use the abbreviation for Australian Football (AF) here?

*Ln315: Why do you use approximately? Is there no exact value for it?

Ln347: Very interesting finding. This reinforce previous articles finding how collective coordination dynamics decrease across the periods of the match, highly influenced by effort accumulation. See:

Duarte, R., Araújo, D., Folgado, H. et al. Capturing complex, non-linear team behaviours during competitive football performance. J Syst Sci Complex 26, 62–72 (2013). https://doi.org/10.1007/s11424-013-2290-3

This may suggest that your used data analysis techniques may be proposed as complementary methods of analysis to approach effort accumulation and acute fatigue effects in collective sports.

Also, your data analysis techniques may be applied to capture congestion not only in competition but also in training when simulating different environments similar to matches. For example, when manipulating player’s space of interaction (Ric et al., 2017) or temporary numerical imbalances (Cantón et al., 2019). Indeed, this study offers objective tools with highly applicability in collective sports.

Ric A, Torrents C, Gonçalves B, Torres-Ronda L, Sampaio J, Hristovski R (2017) Dynamics of tactical behaviour in association football when manipulating players' space of interaction. PLoS ONE 12(7): e0180773. https://doi.org/10.1371/journal.pone.0180773

Canton, A., Torrents, C., Ric, A., Gonçalves, B., Sampaio, J., & Hristovski, R. (2019). Effects of Temporary Numerical Imbalances on Collective Exploratory Behavior of Young and Professional Football Players. Frontiers in psychology, 10, 1968. https://doi.org/10.3389/fpsyg.2019.01968

Ln380: Moreover, this study provides data analysis techniques that take into account the coordination dynamics properties of teams. From a complex systems based-approach it seems more adequate than using isolated and timeless methods (Montull et al., 2022) to assess not only congestion but also, as I mentioned, other sport-related phenomena influencing the collective behaviour of teams as effort, match strategies, numerical imbalances, etc. In this sense, other methods of analysis based on coordination dynamic properties, such as Uncontrolled manifold to assess synergies or network analysis, may help to approach congestion and related phenomena in future research as well.

Montull, L., Slapšinskaitė-Dackevičienė, A., Kiely, J. et al. Integrative Proposals of Sports Monitoring: Subjective Outperforms Objective Monitoring. Sports Med - Open 8, 41 (2022). https://doi.org/10.1186/s40798-022-00432-z

Conclusion:

Ln391: “approximately 25%”, as I mentioned above, cannot be described with its exact value?

6. PLOS authors have the option to publish the peer review history of their article (what does this mean?). If published, this will include your full peer review and any attached files.

Reviewer #1: **Yes: **Heidi Thornton

Reviewer #2: **Yes: **Lluc Montull

---

## [Author Response · Author response to Decision Letter 0]

29 Jun 2022

Reviewer #1: Comments to the authors 

General comments 

Thanks for allowing me to review this manuscript. Overall, the study is of very high quality and the findings are applicable and relevant to those working within AFL. The two methods presented to quantify congestion are novel and highly innovative, requiring extensive analyses so I commend you all on that. A major strength of the study was the ability of the authors to explain the findings in simple, practical terms, useful for practitioners as the analyses would be unfamiliar to most readers. Although I have some understanding of machine learning, much of the analyses were unfamiliar to me, so I can’t comment too much about the appropriateness, however the figures were easy to interpret. The manuscript is very well written, and explains the concept of the study exceptionally, and the as mentioned the findings are explained well. My suggestions/comments are mostly related to some additional detail that I believe is necessary within the methods which are detailed below. Thanks again, and congrats on completing such a high quality manuscript.

Specific comments

Abstract

Line 40: Consider adding ‘method’ between second and aimed. To me it read like an aim of the study, when really you’re referring to the method

Line 40 has been amended to include “method” in between second and aimed.

Introduction

Overall, the introduction is really well written and provides context to the reader about the purpose of the study. 

Line 75-76: Where you refer to the ‘speed’ of the game, you need a bit more detail around this. Specifically, are you referring to m/min (assuming so), but I can imagine many readers may assume speed as in velocity of running and perhaps high-speed running volumes. Provide descriptive data in line 77.

Line 76-77 has been amended to “Preliminary investigations involving the evolution of match-play in AF promoted the notion that the speed of the game was increasing, which was estimated by measuring the average velocity of the ball in m/s.” 

Methods

Line 109: Why was 2020 not included? Was this due to the reduced duration of games?

Yes, the shortened duration of game time made it challenging to compare across seasons. We have included reference to this in lines 111-112 “Matches played in 2020 were not included due to the season alterations that were implemented because of Covid-19.”

Line 112: Did you obtain data from all teams in the AFL or one team? How did you obtain this data if it involved multiple teams? 

Champion Data are the official provider to the AFL and are involved in this study. They collect tracking data on all AFL teams via an existing arrangement. Spatiotemporal data was limited to games played at Marvel Stadium only as it’s the only location that offers LPS technology. 

Line 113: The S5 devices aren’t LPS enabled, these devices would have been worn in the 2019 season, but Vector would have been used in 2020-2021. In 2019, Catapult Clearsky (LPS devices) were worn in Marvel stadium. Can you confirm this information, as well as detail/report any between-device information that’s necessary here.

Yes, the authors have altered the device details to ClearSky. The authors have amended lines 115-117 to update the tracking device details and the corresponding data format (cartesian coordinates) “Positional data in the form of Cartesian coordinates for each match were gathered using Catapult ClearSky 10 Hz local-positioning system (LPS) devices for all 44 participants (Catapult Sports, Melbourne, Australia).” The authors do not have any information on between-device differences. Although, both the S5 and vector units utilise the same ClearSky (LPS) technology. 

Line 133: Please provide detail on your methods of determining player location data. I’m assuming you used GNSS lat/long data, but please include details on this within the methods

Positional data was gathered using LPS technology located at Marvel stadium. Cartesian coordinates were used, which did not require a conversion. The authors have included more information regarding this in lines 115-117 to emphasise LPS technology and cartesian coordinates “Positional data in the form of Cartesian coordinates for each match were gathered using Catapult ClearSky 10 Hz local-positioning system (LPS) devices for all 44 male participants (Catapult Optimeye S5, Catapult Innovations, Melbourne, Australia)”.

Figure 1: The quality of the figure seems quite poor in the PDF. I can see it’s a tif file, but perhaps check it in the next submission. Also, I wonder if it’s necessary to include the top panel here – for simplicity?

We processed the figures using the journal guidelines. We discovered in a previous submission that figures in print return a higher resolution. 

The authors consider the top panel to be necessary to display how the initial cluster labels (top panel) are transformed into the corresponding descriptive labels (bottom panel). We outline these steps in lines 184-188 “Clusters of congestion were originally assigned output labels of 0-n, while -1 was assigned to all points clustered as noise (see Fig 1). These labels where converted to provide a practical description of congestion and to differentiate between separate clusters of congestion if more than one cluster was identified for a unique time interval (see Fig 1).”

Line 193: State where that data is presented – Figure 2 and 3

Line 200 has been amended to include where data is presented “field position (see Fig 2) and quarter (see Fig 3).”

Line 214: Did the same Champion Data staff label/code the disposals?

Yes, the same staff members were used when labelling the training datasets and developing the spatiotemporal features. We have include a reference to this in lines 216-217 “In consultation with the same match analysts from Champion Data.”

Line 233: Should the Shapley Additive exPlanations package be in italics like the others previously reported? 

The authors have altered line 240 to display ‘Shapley Additive exPlanations’ in italics to align with other python packages reported.

Results

Overall, the results explain the findings really well. My only suggestion would be with the figures to make them look a bit ‘cleaner’ and publication worthy. For example, removing the grey section around them, lighten (or remove) the gridlines, position the legend in a consistent spot. Also the colour scheme used across each figure varies quite a bit. I understand the colours are used to represent different things, but perhaps consider using a similar theme

All figures have been amended to remove the grey frame, adjust the gridlines to be more transparent (lightened), and the legend is in a consistent location. The authors maintain that a different colour scheme is helpful when differentiating between the models. 

Discussion

The discussion was well written, and explained the results in simple, applied/practical terms. This is really useful for readers, as the analysis is very complex, the findings need to be interpreted clearly for the translation of this study into practice. 

Line 314: Perhaps link the first two paragraphs together. 

Lines 320-321 have been included to link the first and second paragraph in the discussion has been “This information provides a scalable method to quantify congestion during matches.”

Line 382: Can you state the reasoning for this in the methods 

LPS technology was only available at matches played at Marvel Stadium. Therefore, the analysis was limited to these matches only. We have included more detail explaining the data collection process in lines 112-117 “To ensure consistent tracking data and uniform field dimensions, matches (n = 56) were played at a single stadium (Marvel Stadium, Melbourne, Australia) where the field dimensions were 159.5 m x 128.8 m (length x width). Positional data in the form of Cartesian coordinates for each match were gathered using Catapult ClearSky 10 Hz local-positioning system (LPS) devices for all 44 male participants (Catapult Sports, Melbourne, Australia).”

References

Ref 2, 14, 34, 35 journal title needs to be in capital format

We have formatted these references to include a capital for the journal title

Reviewer #2: The authors are to be commended for a well-written article. It is the first known article that proposes objective data analysis techniques to capture the congestion in Australian football. Both approaches seem to provide more effective information than current methods (i.e., manual input). This not only seems to be useful to evaluate congestion in AF, but also may be practical for many other collective sports-related phenomena. Generally, the topic falls within the scope of the journal and could be of potential interest to its readers.

However, the following minor concerns need to be addressed before publication:

Due to the large amount of abbreviations a list of abbreviations would be recommended, as long as journal’s guidelines approve it, to facilitate the reading. 

We thank the reviewer for the comments. After examining the journal guidelines, it appears a list of abbreviations would not align with the journal guidelines. The authors will gladly oblige the request if the journal allows it. 

Abstract:

Ln34: Australian instead of Australia

Line 34 has been amended to “Australian instead of Australia”

*Ln35: “is an important consideration in identifying passing capacity, assessing fan enjoyment, and evaluating the effect of rule changes”. You mention these aspects as important when studying congestion but they are not mentioned elsewhere in the article aside from the abstract. Why? In my opinion, they should be mentioned if you consider in the introduction or discussion, otherwise I would remove them from the abstract.

The authors have adjusted the discussion to be more explicit in referencing the aspects in the abstract. Lines 342-344 now highlight the impact on passing capacity: “Overall, more than 60% of disposals encountered high congestion or nearby congestion. This suggests that large segments of match-play experiences greater density around the ball-carrier, which may instil pressure and influence passing capacity”. Passing capacity is also referenced in lines 346-348 “Disposals performed under low congestion decreased as teams transitioned the ball towards their attacking end”.

We have amended lines 358-360 to comment on fan enjoyment “In response to a steadily declining scoring rate and a predominantly defensive game style, the AFL, guided by the Laws of the Game Charter, have continually implemented major rule changes to enhance fan enjoyment.”

Rule changes are referred to in lines 366-368 “such rule changes constrain the defending team’s ability to restrict ball transition, thereby allowing for more attacking ball movement for the offensive team, which may increase scoring.”

Ln54: Congestion is already in the title. I would suggest to switch it for another key word to avoid repeat it. In this way it may facilitate this article to be found from more searches.

Line 54 has been amended to remove “Congestion” from the key words

Materials and methods: 

*Ln203 to Ln214: As I see, the definitions of level of congestion and spatiotemporal features for disposal classification model are done by yourself without any evidence but in consultation with professionals. I understand there is no work that establishes previous criteria about it. But why these levels and not others? As you may understand these definitions may be somehow relative to the individual and environmental constraints of the game. For example, for nearby congestion “multiple players with 0-10m of ball-carrier but there is some space to make a decision”: within this 10 m some experienced and fresh player probably will have time to make a clear decision, however, a novice and fatigued young player during a rainy day probably can feel high congestion in this space to make an “adequate” decision. I am not intended to change your definitions but to help future readers to consider different features for classifying the dynamic and nonlinear level of congestion, assessing previously the main constraints of the game (e.g., age of players, level of players, meteorology, etc.). In my opinion, this should be briefly considered as a practical implication (in discussion) for next studies to do not treat it as a universal and fixed rule.

We thank the reviewer for the comments. The authors had similar discussions when formulating the methodology. As you mentioned, due to the paucity of studies investigating congestion, it was difficult to use empirical evidence. As such, we developed the spatiotemporal features with professional analysts at Champion Data due to their have domain expertise. 

We have further included a brief discussion in lines 394-399 that outlines the limitations and future considerations “The machine learning models proposed in this study to quantify congestion were novel, which naturally specifies the parameters used to tune the algorithms were likewise organic. Although the models were thoroughly trialed and tested using various input parameters, a greater implementation from a broader range of experts may assist in ensuring the methodology is valid and reliable and if alterations need to be tailored for specific applications.”

Discussion:

Ln310: Why do you not use the abbreviation for Australian Football (AF) here?

Line 317 has been amended to include the abbreviation “This study developed two methods to measure congestion in AF.” 

*Ln315: Why do you use approximately? Is there no exact value for it?

Each of the three clusters of congestion is separated into four field positions (4) and quarter (4). Thus, we have 24 individual combinations of congestion. When summarising these combinations, the range was between 23% and 26%. This has been included in lines 322-323 “The first method showed that players are within a cluster of congestion (primary or secondary) between 23% and 26% of a typical game”.

Ln347: Very interesting finding. This reinforce previous articles finding how collective coordination dynamics decrease across the periods of the match, highly influenced by effort accumulation. See: Duarte, R., Araújo, D., Folgado, H. et al. Capturing complex, non-linear team behaviours during competitive football performance. J Syst Sci Complex 26, 62–72 (2013). https://doi.org/10.1007/s11424-013-2290-3

This may suggest that your used data analysis techniques may be proposed as complementary methods of analysis to approach effort accumulation and acute fatigue effects in collective sports.

Also, your data analysis techniques may be applied to capture congestion not only in competition but also in training when simulating different environments similar to matches. For example, when manipulating player’s space of interaction (Ric et al., 2017) or temporary numerical imbalances (Cantón et al., 2019). Indeed, this study offers objective tools with highly applicability in collective sports. 

Ric A, Torrents C, Gonçalves B, Torres-Ronda L, Sampaio J, Hristovski R (2017) Dynamics of tactical behaviour in association football when manipulating players' space of interaction. PLoS ONE 12(7): e0180773. https://doi.org/10.1371/journal.pone.0180773

Canton, A., Torrents, C., Ric, A., Gonçalves, B., Sampaio, J., & Hristovski, R. (2019). Effects of Temporary Numerical Imbalances on Collective Exploratory Behavior of Young and Professional Football Players. Frontiers in psychology, 10, 1968. https://doi.org/10.3389/fpsyg.2019.01968

Ln380: Moreover, this study provides data analysis techniques that take into account the coordination dynamics properties of teams. From a complex systems based-approach it seems more adequate than using isolated and timeless methods (Montull et al., 2022) to assess not only congestion but also, as I mentioned, other sport-related phenomena influencing the collective behaviour of teams as effort, match strategies, numerical imbalances, etc. In this sense, other methods of analysis based on coordination dynamic properties, such as Uncontrolled manifold to assess synergies or network analysis, may help to approach congestion and related phenomena in future research as well.

Montull, L., Slapšinskaitė-Dackevičienė, A., Kiely, J. et al. Integrative Proposals of Sports Monitoring: Subjective Outperforms Objective Monitoring. Sports Med - Open 8, 41 (2022). https://doi.org/10.1186/s40798-022-00432-z

Thank you for recommending these articles. The authors discussed player fatigue and how it may influence coordination dynamics. These articles will be very useful in designing future investigations that view congestion within a complex systems framework.

Conclusion:

Ln391: “approximately 25%”, as I mentioned above, cannot be described with its exact value?

Lines 396 have been amended to “The clustering model identified that players were within a cluster of congestion between 23% and 26% of a typical game.”

---

## [Decision Letter · Decision Letter 1]

25 Jul 2022

Quantifying congestion with player tracking data in Australian Football

PONE-D-22-01015R1

Dear Dr. Alexander,

We’re pleased to inform you that your manuscript has been judged scientifically suitable for publication and will be formally accepted for publication once it meets all outstanding technical requirements.

Kind regards,

Gábor Vattay, PhD, DSc

Academic Editor

PLOS ONE

Additional Editor Comments (optional):

Please, before resubmission correct the few typo/missing information mentioned by the reviewer! 

Reviewers' comments:

Reviewer's Responses to Questions

**Comments to the Author**

1. If the authors have adequately addressed your comments raised in a previous round of review and you feel that this manuscript is now acceptable for publication, you may indicate that here to bypass the “Comments to the Author” section, enter your conflict of interest statement in the “Confidential to Editor” section, and submit your "Accept" recommendation.

Reviewer #1: All comments have been addressed

Reviewer #2: All comments have been addressed

2. Is the manuscript technically sound, and do the data support the conclusions?

Reviewer #1: Yes

Reviewer #2: Yes

3. Has the statistical analysis been performed appropriately and rigorously? 

Reviewer #1: Yes

Reviewer #2: Yes

4. Have the authors made all data underlying the findings in their manuscript fully available?

Reviewer #1: Yes

Reviewer #2: Yes

5. Is the manuscript presented in an intelligible fashion and written in standard English?

Reviewer #1: Yes

Reviewer #2: Yes

6. Review Comments to the Author

Reviewer #1: General comments

Great work on making the suggested changes from both reviewers. The manuscript has substantially improved in quality and is very close to being published in my opinion. I have just a few remaining queries and suggestions, that are detailed below.

Specific comments

Abstract: Looks great

Introduction:

Line 66: Consumption of what? Be more specific here or consider using a different word. Reads like consumption of food

Line 77: Please use the correct units of measurement/notation here for m/s, so m·sP-1P

Materials and methods

Line 188: Figure caption is here, but no figure?

Results

The figures look much cleaner now without the gridlines

Discussion

Line 392: Explain what you mean by organic in this context

Reviewer #2: (No Response)

7. PLOS authors have the option to publish the peer review history of their article (what does this mean?). If published, this will include your full peer review and any attached files.

Reviewer #1: **Yes: **Heidi Compton

Reviewer #2: **Yes: **Lluc Montull

---

## [Editor Report · Acceptance letter]

29 Jul 2022

PONE-D-22-01015R1 

Quantifying Congestion with player tracking data in Australian Football 

Dear Dr. Alexander:

I'm pleased to inform you that your manuscript has been deemed suitable for publication in PLOS ONE. Congratulations! Your manuscript is now with our production department. 

Kind regards, 

on behalf of

Dr. Gábor Vattay 

Academic Editor

PLOS ONE